# Osteotropic Effect of Parenteral Obesity in Programmed Male Rats Fed a Calorically Differentiated Diet during Growth and Development

**DOI:** 10.3390/ani12182314

**Published:** 2022-09-06

**Authors:** Radoslaw Piotr Radzki, Marek Bienko, Dariusz Wolski, Pawel Polak, Kinga Topolska, Mateusz Wereszczynski

**Affiliations:** 1Department of Animal Physiology, Faculty of Veterinary Medicine, University of Life Sciences in Lublin, Akademicka 12, 20-033 Lublin, Poland; 2St. Johns’ Oncology Center in Lublin (COZL) Trauma and Orthopaedic Surgery Department, Jaczewskiego 7, 20-090 Lublin, Poland; 3Department of Plant Product Technology and Nutrition Hygiene, Faculty of Food Technology, University of Agriculture in Krakow, al. Mickiewicza 21, 31-120 Krakow, Poland; 4Chair and Department of Rehabilitation and Orthopaedics, Jaczewskiego 8, 20-090 Lublin, Poland

**Keywords:** obesity, nutrition, bone metabolism, prenatal programming, offspring, growing, peripheral quantitative computed tomography (pQCT), dual X-ray absorptiometry (DXA), mechanics and strength of bones

## Abstract

**Simple Summary:**

Parental obesity affects skeletal metabolism in offspring. This relationship is called “nutritional programming”. During the weaning period, they are more highly mineralized and mechanically resistant. It was interesting for us whether changing or continuing the feeding of male offspring with a standard or high-energy diet may have different metabolic effects on bone tissue. Our previous studies on females have shown that the beneficial direction of change is the replacement of the standard diet with a high-energy diet; the reduction of the caloric content of food (change from a high-energy to a properly balanced diet) leads to disorders of skeletal growth and development. In males, any change in diet inhibited skeletal development, and the bones were weaker. The most effective was the continuation of high-energy nutrition, which, in males at 49 and 90 days of age, was manifested by stronger bones. This proves that males and females react differently to a change in the caloric content of the diet during the period of growth and development.

**Abstract:**

The experiment was undertaken to assess whether the continuation or change of the parents’ diet affects the previously programmed bone metabolism of the male offspring during its growth and development. A total of 16 male and 32 female Wistar rats were divided into groups and fed a standard (diet S) or high-energy (diet F). After the induction of obesity, the rats from groups S and F, as the parent generation, were used to obtain male offspring, which were kept with their mothers until the weaning day (21 days of age). In our earlier study, we documented the programming effects of the diet used in parents on the skeletal system of offspring measured on the weaning day. Weaned male offspring constitute one control group—parents and offspring fed the S diet. There were three experimental groups, where: parents received diet S and offspring were fed with the F diet; parents were treated with the diet F, while offspring received the S diet; and parents and offspring were fed with the diet F. The analyses were performed at 49 and 90 days of life. After sacrifice, cleaned-off soft tissue femora were assessed using peripheral quantitative computed tomography (pQCT), dual X-ray absorptiometry (DXA), and a three-point bending test. We observed that changing and continuation of nutrition, applied previously in parents, significantly influenced the metabolism of the bone tissue in male offspring, and the osteotropic effects differed, depending on the character of the nutrition modification and age. Additionally, an important conclusion of our study, regarding the previous, is that nutrition modification, affecting the metabolism of bone tissue, also depends on the sex.

## 1. Introduction

Obesity and overweight in men are serious medical problems. According to World Health Organization data from 2016, approximately 11.1% of all men over 18 were obese and as much as 38.5% overweight [1]. For women, the proportion is 15.1% and 39.2%, respectively. More detailed WHO observations indicate that in Japan, Korea, China, Germany, France, the United Kingdom, and the USA, obesity was more often diagnosed among men [1]. Obesity is also a growing problem in children. About 340 million children aged 5–19 are struggling with the problem of overweight or obesity [2].

“Programming” is the term used to describe the early (in utero or early childhood) factors that influence growth and metabolism, in the context of their ongoing effects on tissue and organ function and, thus, the risk of disease in later life [3]. This relationship shows that the quality and quantity of nutrition during the perinatal periods may modulate the genetic information originating from the parents.

The opinion on the influence of obesity on the metabolism of bone tissue is ambiguous, and the two views clash. In 1998, Nguyen et al. [4], based on retrospective studies, showed that body weight and bone mass are subject to strict genetic control. Genetic conditions determine nearly 80% of bone peak mass, while 20% depend on the environmental influences [5]. In addition to genetic factors, it is believed that the increased mechanical loads resulting from obesity and overweight have an osteoprotective effect by inhibiting osteoblast apoptosis and stimulating the formation of bone tissue [6,7]. Leptin may also indicate a favorable relationship between the adipose and bone tissue. It is assumed that an increase in the level of this adipokinin is associated with an increase in bone mineral density (BMD) [8,9]. Currently, the results of many studies form the basis of the theory that there is competition between the fat and bone-forming cells and increased levels of adipose tissue can damage bones, often accelerating the process of bone loss [10,11]. Obesity promotes the development of systemic inflammation, which is manifested by an increase in the concentration of pro-inflammatory cytokines, which cause increased bone resorption [12,13].

The programming influence of parental obesity on the bone metabolism of the offspring is a fact [14]. In earliest studies, we described the osteotropic effects of a standard and high-energy diet in different variants in pre-programmed female offspring in the next stages of life [15]. However, Saxon et al. [16] documented the existence of differences in the development of the skeletal system of growing females and males. They concern mineralization and structural features that have a significant impact on mechanical strength. Therefore, we hypothesized that changing or continuing the feeding of male offspring with a standard or high-energy diet may have different metabolic effects on bone tissue than in females. The aim of the study was to determine, in pre-programmed nutritionally male offspring, the osteotropic effect of continuing the eating habits of parents in the use of a high-energy or standard diet. The influence of the change in the offspring, as well as the diet used in the parents from standard to high-energy and vice versa, were also assessed. These studies were carried out in key developmental periods, i.e., in the period of sexual maturity (49 days of age) and reproductive maturity (90 days of age).

## 2. Materials and Methods

### 2.1. Animal Procedures

The study was conducted using Wistar rats (32 female and 16 male), which constituted the parental generation. Female and male rats of the parental generation made it possible to obtain offspring used in the main part of the experiment (Figure 1). For this purpose, the rats were randomly divided (16 females and 8 males) into groups of females and males and fed a high-energy diet (F) (17.6 MJ/kg—(16% fat, 65% carbohydrates, and 19% protein) for induction obesity and control groups of both sexes receiving the standard diet (S) (11.5 MJ/kg) (Agropol, Motycz, Poland) (for details see Appendix A). Diet S and F were applied for 90 days, i.e., the time necessary to induce obesity in rats treated with diet F (Appendix A). 

In the next step, a high-energy diet in group F was replaced by a standard to eliminate the possibilities affecting high-energy diet F during fertilization and gestation. Then, females and males receiving the F or S diet were placed in specially prepared cages in the proportion of 2 females/1 male for fertilization. After 14 days, the females were separated and placed in separate cages. The average litter did not exceed 8–12 newborns. This part of the experiment aimed to demonstrate the existence of a programming effect of parental obesity on the metabolism of the skeletal system of the female and male offspring tested at weaning. The results of these studies were described in our earlier work [14]. On the 21st day (the weaning day) after birth, the rats were sexed, and males were randomly divided into one control and three experimental groups, based on the information presented below.

Control (S/S)—parents fed S diet—male offspring treated with the S diet (*n* = 16);Group (S/F)—parents fed the S diet—male offspring treated with the F diet (*n* = 16);Group (F/S)—parents fed the F diet—male offspring treated with the S diet (*n* = 16);Group (F/F)—parents fed the F diet—male offspring treated with the F diet (*n* = 16).

The rats had free access to diet and water, and body weight was measured once a week (Figure 1). The blood serums were collected and frozen at −30 ℃ for further biochemical analysis. The femora were carefully cleaned of soft tissues for densitometric, tomographic, and mechanical analyses.

### 2.2. Mechanical Analysis of the Femur

Mechanical parameters, of isolated femora, such as maximum bone strength (F_max_), Young modulus of elasticity (E_mod_), and work to maximum strength index (W/F_max_), were investigated with the use of a 3-point bending test performed by ZwickRoell Z10 (Zwick GmbH and Co. KG, Ulm, Germany) apparatus, equipped with measuring head (Xforce HP series), working from 0–1 kN range. The bone samples, analyzed as tube model, were estimated by the testXpert II v3.1 ZwickGmbH & Co. KG (Ulm, Germany) software [17,18].

### 2.3. Dual X-ray Absorptiometry (DXA) Measurements of the Total Skeleton and Isolated Tibiae

Densitometry analysis of total skeleton (planar bone mineral density Ts.BMD, bone mineral content Ts.BMC and area Ts.Ar), isolated femora (f.BMD; f.BMC; f.Ar), and total body composition (lean mass LM, fat mass FM) were performed with the use Norland Excell Plus Densitometer and Illuminatus Small Subject Scan v.4.7.2 software (Norland, Fort Atkinson, WI, USA). After the scout scan, the region of interest (ROI) was defined manually by the operator, and a measured scan was performed [19].

### 2.4. Peripheral Quantitative Computed Tomography (pQCT)

Bone samples were analyzed using a pQCT Stratec XCT Research SA Plus under control of Stratec software, version 6.20 C (Stratec Medizintechnik GmbH, Pforzheim, Germany). The trabecular and cortical compartments of isolated femora were measured in the distal metaphysis and middle of the shaft of column, respectively, as described previously [14,20,21]. 

### 2.5. Biochemical Analysis

The serum concentration of osteocalcin (OC) (cat. No. AC-12F1—Immunodiagnostic Systems, Bolton, UK), activity of bone-fraction of alkaline phosphatase (bALP) (cat. No. E-EL-R1109—Immunodiagnostic Systems, Bolton, UK), and level of C-terminal telopeptides of type I collagen (CTX-I) (cat. No. AC-06F1—Immunodiagnostic Systems, Bolton, UK) were established using ELISA method. The serum level of ionized calcium (cat no W6504) and phosphorus (cat. No. F6516-100) were performed spectrophotometrically (Alphadiagnostic, Warszawa, Poland).

### 2.6. Statistical Analysis

The statistical analyzes were made with the use of STATISTICA software v. 13.0 (TIBCO Software Inc. Palo Alto, CA, USA). Using the Shapiro–Wilk test, the normality of the data distribution was established. Then, the multivariate comparisons were ascertained by ANOVA. Where the ANOVA test was statistically significant (*p* < 0.05), the differences between groups were established by applying the post hoc Tukey test. The results were presented as the mean value and standard error of the mean (x ± SEM). The statistically significance *p* values were identified at *p* < 0.05.

## 3. Results

### 3.1. Mechanical Strength

Changes in the offspring the parents’ eating habits onto diet F (S/F group) or its continuation (F/F group) significantly reduced the E_mod_ at 49 days (*p* < 0.05), with the lowest values in the F/S groups (Figure 2). At 90 days of age, E_mod_ in groups F/S and S/F were significantly lower (vs. S/S group), with similar values of this parameter in the F/F group. Similar characteristics of changes concerned F_max_ and W/F_max_ of the femora (Figure 2).

### 3.2. DXA Analysis of Femur

Statistically, the lowest Ts.BMD, Ts.BMC, and Ts.Ar were found in 49 and 90 days old F/S males (*p* < 0.05) (Table 1). Additionally, the density and mineral content, as well as, the femoral area (f.BMD, f.BMC, and f.Ar) were significantly the lowest in the F/S offspring in both studied age groups (*p* < 0.05). Significantly lower values of BMC, BMD, and Ar of the total skeleton and isolated femurs were also noted in the F/F group of 49-day-old rats, while, on the 90-day, the values of the parameters assessed did not differ from those found in the control group (S/S). The use of diet F in the S/F group, in most cases, did not significantly affect the densitometric parameters of the skeleton and femur of 49- and 90-day-old male rats, showing only a tendency to lower values vs. S/S (Table 1). 

### 3.3. DXA Analysis of Body Composition and Body Weight

DXA analysis of body composition on day 49 showed significantly lower values of soft tissue, fat mass, and lean mass in all experimental groups, compared to the S/S control (*p* < 0.05). On the 90 days, the highest values of these parameters were found in F/F males, with the statistical confirmation of the differences of the mean values in the case of soft tissue (*p* = 0.02) and fat mass (*p* = 0.002) (Table 2). The 49-day-old rats in the S/S group (*p* < 0.5) had the significantly highest body weight, while the F/F males (*p* < 0.05) had the highest body weight at 90 days of age. The significantly lowest body weight was found in the 49- and 90-day-old rats in the F/S group (*p* < 0.05) (Table 2).

### 3.4. pQCT Analysis—Trabecular Bone Tissue in the Proximal Metaphysis

Changing the diet in the offspring to standard (group (F/S) or high-energy (group S/F), as well as the continuation of high-energy nutrition (group F/F), resulted in a significant reduction of Tb.BMC and Tb.vBMD of the male femur at 49 days of age (*p* < 0.05). The Tb.Ar of the examined bones was significantly lower only in the F/S group (*p* = 0.006). A similar relationship was noted at 90 days. However, Tb.vBMD and Tb.Ar were lower by 10% and 5%, respectively (Table 3).

### 3.5. pQCT Analysis—Cortical Bone Tissue in the Mid-Shaft Diaphysis

The change of the diet (groups (F/S and S/F) and continuation of the high-energy diet brought about the outcome that Ct.BMC, Ct.vBMD, and Ct.Ar, as measured in the mid-shaft of the femur diaphysis, were, in most cases, significantly lower (*p* < 0.05) in males at 49 days of age, with the lowest values of the analyzed parameters in the F/S group. At 90 days of age, significantly lower values of Ct.BMC, Ct.vBMD, and Ct.Ar were recorded only in the F/S group (*p* < 0.05) (Table 4). The values of the densitometric parameters and cortical bone area in the S/F and F/F groups did not show significant differences, as compared to the S/S control. However, the Ct.BMC and Ct.Ar of the femur of the F/F males were higher by 5% and 6%, respectively. Significantly lowest PERI.C were found in the mid-shaft of the femur in the F/S group at 49 (*p* = 0.0002) and 90 days of age, with similar values of this parameter in the other experimental groups (vs. the S/S group). The lowest values of endocortical circumferences (ENDO.C) were found in the group of 49-day-old males of F/S (6%), and the highest in the group of F/F (*p* = 0.011). Moreover, 5% higher ENDO.C values were recorded in the femur of males in the S/F group. At 90 days of age, the ENDO.C values of the femurs of all groups did not differ from each other. The femora of 49 and 90-day-old males in the F/S groups (*p* < 0.05) were characterized by the lowest cortical thickness (Ct.Th). Significantly lower Ct.Th values were recorded in the groups of 49-day-old S/F and F/F males. Ct.Th of the examined bones of the remaining groups of male offspring at 90 days of age were at a similar level as in the S/S group (Table 4).

### 3.6. Biochemical Analysis

The lowest concentration of osteocalcin (OC) was found in the blood plasma of males in the F/S group at 49 (*p* = 0.0002 vs. S/S) and 90 days of life (*p* = 0.005 vs. S/S). A lower concentration of OC was also found in S/F males at the age of 49 (*p* = 0.01) and 90 days, as compared to S/S_49 and S/S_90, respectively. The concentration of OC in the F/F group was similar to the S/S in both age groups (Table 5). The highest concentration of CTX-I was recorded in 49 and 90-day-old males in the F/S group (*p* < 0.05). In addition, a higher concentration of CTX-I was found in both age groups of S/F males (*p* < 0.05). Continued use of diet F did not significantly affect the level of CTX-I in males at 49 days of age, while, on the 90th day, its value was significantly lower (*p* = 0.007 vs. S/S) (Table 5). Change of the diet (groups F/S and S/F) significantly increased bALP activity at 49 and 90 days of age. The activity of bALP in 49-day-old male F/F was similar to the values recorded in the control, while, at 90 days of age, the original value significantly increased by 31% (*p* = 0.0004). The lowest Ca level was noted in 49-day-old F/S rats (*p* = 0.0001), while the values in the other groups of rats at this age were on a similar level. On the 90th day, the lowest Ca concentration was in the F/S group (*p* = 0.0001). A significant decrease in Ca concentration was also recorded in the S/F group (*p* = 0.02), while, in the F/F rats, the calcium level was on the similar level as in the control (Table 5).

## 4. Discussion

Our previous studies have shown that parental obesity has a programming influence on skeletal metabolic processes in the offspring [14]. We also demonstrated that, on the day of weaning, the tibia of the female and male offspring, coming from obese parent, is characterized by stronger growth and more intense mineralization and, thus, greater mechanical endurance. Parents’ eating habits are very often continued by their offspring in the next stages of life, which, in the case of obesity, exacerbates the problem of maintaining healthy body weight, initiated already in the perinatal period [22]. It would seem that this dependence applies only to people. However, we should remember that nutrition mistakes are also noted in animal husbandry, which results from too high a food dose, too high energy value of the diet, and the low physical activity of animals [23]. In this case, the fault lies solely with the breeder, but the animals suffer [24]. Obesity, which affects both humans and animals, is at the root of numerous dietary programs to lose weight [25]. Their exclusive focus on weight reduction, often, especially in the period of growth and development, does not take into account that changes to the previously used high-energy nutrition to a diet with normal or reduced energy value may affect organogenesis, including the skeletal system. Unfortunately, a properly balanced diet is often replaced with high-energy nutrition. Additionally, in this case, the skeletal system reacts by changing its metabolism [15]. Any changes that take place in the bone tissue, related to its mineralization or structure, has a direct impact on the mechanical strength of bones [26]. In a young developing organism, the mechanical strength is constantly modified, adapting to the changing mechanical loads resulting from physical activity. The diet followed during development is also important [27]. In our study, changing the diet used in previously nutritionally programmed male offspring (groups S/F and F/S), as well as the continuation of a high-energy diet (group F/F), markedly decreased the strength of the femurs at 49 days of age. As in the previously cited studies, in females [15], as well as in males, the bones in the F/S group were characterized by the lowest mechanical strength. However, it is surprising that there was a significant decrease in bone mechanical strength in the group in which the standard diet on the day of weaning was replaced with a high-energy diet (S/F group). In females, such a change of diet did not affect the mechanical parameters of the femurs at 49 days of age. The treatment with a high-energy diet (S/F and F/F groups) between 49 to 90 days of age compensated for the earlier lower values of mechanical endurance, which, at 90 days of age, were similar to the S/S control group. In the same period of female life, the introduction of the F diet on the day of weaning (S/F group) significantly increased the strength of the femur, in relation to the control. The increased of the mechanical strength of male’s femur, fed a high-energy diet, was also observed by Nunes et al. [28] in rats and Lanham et al. [29] in mice. In contrast, Bielohuby et al. [30] noted the decrease of the tibia strength in male rats; however, the authors used the diet with a high content of fat (66% and 94.5%) and very low (1% and 1.3%) level of carbohydrates. Replacing the F diet used in parents with a S diet in male offspring, between 21 and 90 days of age, resulted in significantly lower mechanical endurance. Similar effects were also observed in previously studied females [15].

The strength of bone depends on numerous factors. The level of mineralization, size, microstructure, and shape of the hydroxyapatite crystals, as well as the spatial distribution of the collagen fibers, are of key importance here [26]. The strength analysis, based on the force acting perpendicular to the bone long axis, was applied in the mid-shaft. The cortical compartment, which dominates in this part, is mainly responsible for the resistance of the bone [31]. Tot.BMC and Ct.BMC, analyzed in the mid-length of the femurs of 49-day-old rats, showed a significant reduction in the values in the groups of changing the diet previously used by the parents and in the group of continuing the F diet. The change of nutrition also inhibited radial bone growth, which resulted in a lower area of the total cross-sectional and surface of the compact bone tissue. The consequence of these changes was the lower volumetric mineral density and mineral content (Tot.vBMD and Ct.vBMD). At 90 days of age, these changes were more or less compensated for in the individual groups, especially in the F/F group. The use of the F diet in the F/F group eliminated the differences in densitometric parameters, compared to the S/S control group. Additionally, the mineralization of the male bones in the S/F group was similar. On the other hand, the change from F to S diet was still manifested by the lowest mineralization.

Nutritional modifications in male offspring (groups F/S and S/F), as well as males in the F/F group, also showed changes in bone geometric features. These consisted of a reduction in the thickness of the cortical bone. The strongest influence of nutrition on the developmental changes in the femurs was evident in males in the group in which the high-energy diet was replaced with the standard diet (F/S). Such changes in nutrition triggered the strongest inhibition of mineralization and disorders of bone architecture in both age groups. Although both age groups of males receiving diet F (groups S/F and F/F) showed a disturbance in the structure of the cortical bone tissue on day 49, which was reflected in the smaller thickness and surface area of the cortical compartment, on day 90, the values of these parameters did not differ from those recorded in the S/S group. The disorders resulting from changing from a high-energy diet to a standard diet also brought about a significant inhibition of bone length. Diet modifications and the continuation of high-energy nutrition based on the F diet, moreover, had a strong effect on the metabolism of trabecular bone tissue. It should be emphasized that this type of bone tissue is more susceptible to multidirectional influences, such as hormonal disorders, environmental factors, or nutritional influences; its metabolism is more intense, and changes are more strongly expressed than in the cortical compartment [32]. Our data are in line with the observation described by Costa et al. [33] on the model of male rats.

Tomographic (pQCT) analysis of bone tissue is characterized by high accuracy, and the used algorithm enables a separate analysis of cortical and trabecular bone tissue in a 3D system. Nevertheless, densitometry (DXA) testing remains the “gold standard” for the assessment of bone tissue. This method allows for the examination of bones in a 2D system. Additionally, the analysis may cover the entirety of the skeleton or isolated bones. Observing the changes in the mineral content of the whole skeleton (Ts.BMC) and mineral density (Ts.BMD), as well as the femur (f.BMC and f.BMD), an inhibition of mineralization is visible in the offspring at 49 days of age, whose diet was changed, in relation to the one received by the parents so far (groups F/S and S/F). A similar inhibitory effect was found in males from obese parents who continued the F diet. Interestingly, the use of diet F in males from obese parents (group F/F) in the period from 49 to 90 days of age resulted in the equalization of the mineral content of both the whole skeleton (Ts.BMC) and femur (f.BMC), the values of which were similar or even higher, as in the S/S control group. Nunes et al. [28] also observed an increase in femur mineralization, in response to a high-energy diet.

Moreover, changing from a high-energy diet to a standard diet (F/S group) during male growth and development inhibits the growth of the skeleton and isolated bone. On the 49th day, the total skeleton area was 28% smaller, and the area of the isolated femur was 21% smaller than in the S/S group. On the 90th day, these differences were less and amounted to 17% and 10%, respectively, which indicates that this direction of changes in nutrition is not recommended during the period of intensive organism growth. Additionally, the use of the F diet in males in the S/F and F/F groups limited skeletal development, but only between 21 and 49 days of life. In contrast, dieting in F in these groups in the period from 49 to 90 days of age eliminated these differences, and the total area of the skeleton and femoral area in these groups did not differ from the control (S/S group). Changes in Ts.BMD and f.BMD, as well as the area of the entire skeleton and isolated femur, are the consequence of changes in Ts.BMD and f.BMD. The changes noted in the group of 90-day-old F/F males are particularly surprising because the total skeletal and femoral BMDs were higher than those in the S/S control group.

## 5. Conclusions

Despite the prevailing opinions that overweight and obesity are factors contributing to the degradation of bone tissue [34], it seems that the role of adipose tissue in the metabolism of bone tissue depends on the age at which such a relationship is considered. The results of our research indicate that, during the growth and development of male rats, the use of a high-energy diet had a beneficial effect on the mineralization and structure of bone tissue, and, consequently, its mechanical properties. Thus, during the period of growth and development, the effect of a high-energy diet appears to be a factor promoting the development of the skeletal system; after completion of growth, its effects are contradictory. This thesis seems to be confirmed by the studies of Papageorgiou et al. [35]. On the other hand, changing the diet from the parent’s diet to the F or S diet inhibits the growth and development of the femurs in the male offspring. It should be emphasized that a high-energy diet (F) applied from 49 to 90 days of age allows for the compensation of the functional state of the bone tissue, and its mineralization and mechanical strength do not differ from the control group. The use of the S diet in the same period did not improve the quality of bone tissue, and the femurs were characterized by lower mechanical strength. What is more, it should be underlined that there is a visible difference in the reaction of the skeletal system of females [15] and males, who, on the day of weaning, changed their current S diet to the F diet. This demonstrates that females and males differently respond to the increase in the energy value of the diet followed during the period of growth and development. In females, this change increased bone mineralization and mechanical strength, while, in males, the reaction was the opposite.

## Figures and Tables

**Figure 1 animals-12-02314-f001:**
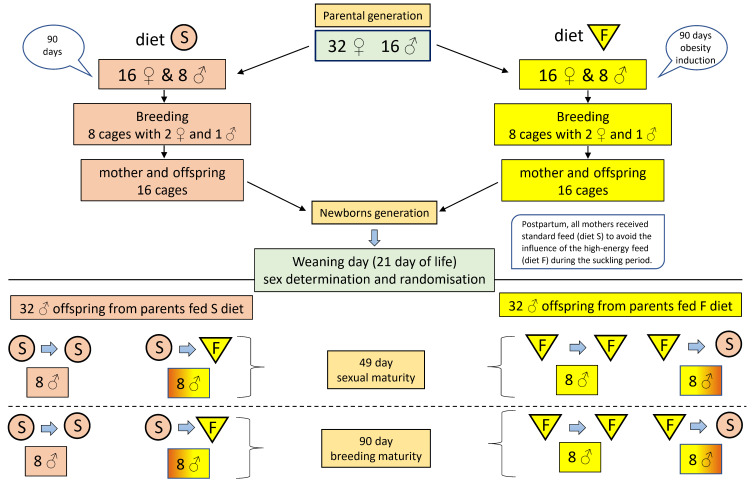
Experimental design. F—high-energy diet; S—standard diet.

**Figure 2 animals-12-02314-f002:**
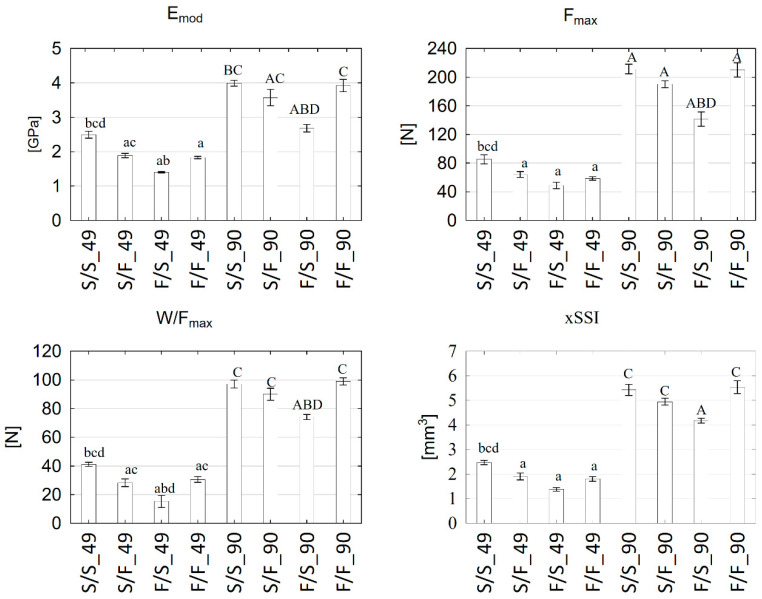
The mechanical parameters of the femur of male rats on the 49 and 90 days of age. Explanations: results are the means ± SEM (*n* = 8). a vs. S/S_49; b vs. S/F_49; c vs. F/S_49; d vs. F/F_49; A vs. S/S_90; B vs. S/F_90; C vs. F/S_90; D vs. F/F_90.

**Table 1 animals-12-02314-t001:** DXA parameters of the entire skeleton and femur.

Item	S/S 49	S/F 49	F/S 49	F/F 49	S/S 90	S/F 90	F/S 90	F/F 90
Densitometry of whole skeleton								
Ts.BMD (g/cm^2^)	0.108 ± 0.003 ^bcd^	0.096 ± 0.003 ^a^	0.093 ± 0.003 ^a^	0.092 ± 0.001 ^a^	0.151 ± 0.001 ^BC^	0.142 ± 0.001 ^AD^	0.134 ± 0.001 ^AD^	0.159 ± 0.003 ^BC^
Ts.BMC (g)	5.32 ± 0.49 ^cd^	4.50 ± 0.48	3.15 ± 0.27 ^a^	4.20 ± 0.18 ^a^	13.3 ± 0.06 ^C^	12.3 ± 0.1 ^C^	10.6 ± 0.5 ^ABD^	13.1 0.5 ^C^
Ts.Ar (mm^2^)	52.0 ± 1.7 ^cd^	46.1 ± 4.1 ^c^	37.3 ± 2.6 ^abd^	45.2 ± 1.3 ^ac^	91.3 ± 1.26 ^C^	86.6 ± 1.3 ^C^	75.8 ± 2.2 ^ABD^	86.7 ± 1.0 ^C^
Densitometry of isolated femur								
f.BMD (g/cm^2^)	0.075 ± 0.003 ^cd^	0.071 ± 0.003 ^c^	0.058 ± 0.002 ^ab^	0.064 ± 0.001 ^a^	0.134 ± 0.002 ^D^	0.124 ± 0.002 ^CD^	0.116 ± 0.002 ^BD^	0.113 ± 0.002 ^ABC^
f.BMC (g)	0.206 ± 0.016 ^cd^	0.184 ± 0.015 ^cd^	0.123 ± 0.005 ^abd^	0.158 ± 0.006 ^abc^	0.503 ± 0.007 ^CD^	0.460 ± 0.011 ^CD^	0.407 ± 0.003 ^ABD^	0.498 ± 0.014 ^ABC^
f.Ar (mm^2^)	2.89 ± 0.09 ^cd^	2.56 ± 0.10 ^c^	2.12 ± 0.07 ^abd^	2.45 ± 0.05 ^ac^	3.90 ± 0.07 ^D^	3.68 ± 0.06 ^D^	3.52 ± 0.06 ^D^	3.95 ± 0.09 ^ABC^

Explanations: results are the means ± SEM (*n* = 8). a vs. S/S_49; b vs. S/F_49; c vs. F/S_49; d vs. F/F_49; A vs. S/S_90; B vs. S/F_90; C vs. F/S_90; D vs. F/F_90.

**Table 2 animals-12-02314-t002:** Body weight and parameters of body composition.

Item	S/S 49	S/F 49	F/S 49	F/F 49	S/S 90	S/F 90	F/S 90	F/F 90
Body weight (g)	302.00 ± 23.2 ^bcd^	234.42 ±13.41 ^ac^	192.43 ± 7.96 ^abd^	236.23 ± 8.73 ^ac^	549.75 ± 9.67 ^BCD^	509.55 ± 14.68 ^AD^	496.67 ± 11.12 ^AD^	620.50 ± 8.76 ^ABC^
Soft tissue (g)	278.6 ± 22.2 ^bcd^	214.9 ± 19.1 ^a^	173.6 ± 8.2 ^a^	212.3 ± 8.4 ^a^	518.7 ± 4.7 ^CD^	477.0 ± 14.9 ^D^	459.5 ± 2.6 ^AD^	587.7 ± 12.0 ^ABC^
Fat mass (g)	16.9 ± 2.2 ^c^	19.7 ± 2.5	11.4 ± 2.9 ^a^	17.4 ± 3.0	75.9 ± 4.8 ^D^	81.5 ± 5.1 ^D^	72.9 ± 4.1 ^D^	101.7 ± 2.8 ^ABC^
Lean mass (g)	254.4 ± 14.9 ^bcd^	198.8 ± 15.8 ^a^	171.3 ± 4.8 ^a^	194.9 ± 5.9 ^a^	449.8 ± 7.2 ^BC^	395.6 ± 13.2 ^AD^	397.5 ± 14.0 ^AD^	486.1 ±18.1 ^BC^

Explanations: as in Table 1.

**Table 3 animals-12-02314-t003:** Tomographic (pQCT) analysis in the distal femur metaphysis.

Item	S/S 49	S/F 49	F/S 49	F/F 49	S/S 90	S/F 90	F/S 90	F/F 90
Total cross-section parameters								
Tot.BMC (mg/mm)	9.7 ± 0.3 ^bcd^	8.4 ± 0.2 ^a^	6.9 ± 0.15 ^a^	7.8 ± 0.2 ^a^	17.5 ± 0.4 ^BCD^	14.3 ± 0.6 ^ACD^	12.9 ± 0.2 ^ABD^	16.2 ± 0.2 ^ABC^
Tot.vBMD (mg.mm^3^)	505.8 ± 8.7 ^bcd^	475.4 ± 8.5 ^ac^	435.8 ± 9.3 ^abd^	475.0 ± 6.9 ^ac^	644.4 ± 11.6 ^BC^	616.6 ± 11.0 ^AC^	574.0 ± 13.2 ^ABD^	625.1 ± 16.5 ^C^
Tot.Ar (mm^2^)	19.1 ± 0.8 ^c^	17.9 ± 0.9 ^c^	14.5 ± 0.6 ^abd^	16.2 ± 0.7 ^c^	27.2 ± 0.8 ^BC^	23.4 ± 0.8 ^AD^	21.1 ± 0.9 ^AD^	25.7 ± 0.8 ^BC^
Trabecular bone tissue parameters								
Tb.BMC (mg.mm)	2.8 ± 0.2 ^bcd^	2.0 ± 0.1 ^a^	1.6 ± 0.1 ^a^	2.0 ± 0.1 ^a^	5.6 ± 0.2 ^BCD^	3.8 ± 0.1 ^ACD^	3.3 ± 0.1 ^ABD^	4.4 ± 0.1 ^ABC^
Tb.vBMD (mg.mm^3^)	318.8 ± 5.1 ^bcd^	278.9 ± 5.1 ^acd^	200.4 ± 5.1 ^abd^	246.9 ± 7.8 ^abc^	457.5 ± 8.2 ^BCD^	318.1 ± 6.3 ^A^	358.37 ± 12.2 ^A^	422.6 ± 10.5 ^A^
Tb.Ar (mm^2^)	8.6 ± 0.4 ^c^	8.1 ± 0.4 ^c^	6. ± 0.3 ^abd^	7.3 ± 0.3 ^c^	12.2 ± 0.4 ^BC^	10.5 ± 0.4 ^AD^	9.5 ± 0.4 ^AD^	11.6 ± 0.4 ^BC^

Explanations: as in Table 1.

**Table 4 animals-12-02314-t004:** Tomographic (pQCT) parameters of the femur in the midshaft diaphysis.

Item	S/S 49	S/F 49	F/S 49	F/F 49	S/S 90	S/F 90	F/S 90	F/F 90
Total cross-section parameters								
Tot.BMC (mg/mm)	6.0 ± 0.3 ^bcd^	5.3 ± 0.1	4.0 ± 0.2 ^a^	4.8 ± 0.1 ^a^	11.2 ± 0.1 ^C^	10.7 ± 0.2 ^C^	9.9 ± 0.2 ^ABD^	11.2 ± 0.1 ^C^
Tot.vBMD (mg/mm^3^)	678.6 ± 6.9 ^cd^	630.2 ± 20.9 ^c^	513.6 ± 8.7 ^ab^	582.9 ± 14.2 ^a^	889.5 ± 14.5	844.9 ± 12.6	776.8 ± 12.4 ^D^	893.2 ± 5.8 ^C^
Tot.Ar (mm^2^)	8.7 ± 0.1 ^c^	8.0 ± 0.1 ^c^	6.5 ± 0.3 ^abd^	7.7 ± 0.2 ^c^	12.5 ± 0.2 ^BC^	11.9 ± 0.2 ^C^	11.5 ± 0.3 ^ABD^	13.0 ± 0.2 ^C^
Cortical bone tissue parameters								
Ct.BMC (mg/mm)	5.0 ± 0.1 ^bcd^	4.4 ± 0.1 ^ac^	3.6 ± 0.1 ^abd^	4.0 ± 0.1 ^ac^	10.0 ± 0.2 ^C^	9.6 ± 0.1 ^C^	8.7 ± 0.1 ^ABD^	10.5 ± 0.1 ^C^
Ct.vBMD (mg/mm^3^)	1244.8 ± 9.0 ^bc^	1187.9 ± 10.2 ^a^	1150.0 ± 11.0 ^ad^	1195.5 ± 6.9 ^c^	1389.9 ± 8.6 ^C^	1380.8 ± 5.2 ^C^	1335.7 ± 9.7 ^AB^	1369.3 ± 15.6
Ct.Ar (mm^2^)	4.3 ±0.1 ^bcd^	3.7 ± 0.1 ^ac^	3.0 ± 0.1 ^abd^	3.5 ± 0.1 ^ac^	7.2 ± 0.1 ^CD^	6.9 ± 0.1 ^CD^	6.5 ± 0.1 ^ABD^	7.7 ± 0.1 ^ABC^
Ct.Th (mm)	0.5 ± 0.01 ^bcd^	0.4 ± 0.01 ^acd^	0.4 ± 0.01 ^ab^	0.4 ± 0.01 ^ab^	0.7 ± 0.01 ^C^	0.7 ± 0.01 ^C^	0.6 ± 0.01 ^AB^	0.7 ± 0.^01^
Peri.C (mm)	10.2 ± 0.2 ^c^	10.1 ± 0.1 ^c^	9.1 ± 0.1 ^abd^	9.8 ± 0.1 ^c^	12.5 ± 0.2	12.2 ± 0.2 ^D^	12.0 ± 0.1 ^D^	12.8 ± 0.1 ^BC^
Endo.C (mm)	7.1 ± 0.1	7.4 ± 0.1 ^c^	6.6 ± 0.1 ^bd^	7.5 ± 0.1 c	8.1 ± 0.1	7.9 ± 0.2	7.9 ± 0.1	8.2 ± 0.2

Explanations: as in Table 1.

**Table 5 animals-12-02314-t005:** The levels of the markers of bone metabolism, calcium, and phosphorus concentration.

Item	S/S 49	S/F 49	F/S 49	F/F 49	S/S 90	S/F 90	F/S 90	F/F 90
ALP (U/L)	432.0 ± 12.1 ^bc^	359.1 ± 13.4 ^ad^	329.9 ± 17.1 ^ad^	445.0 ± 15.1 ^bc^	203.0 ± 15.0 ^D^	179.0 ± 17.5 ^D^	188.3 ± 14.9 ^D^	306.6 ± 29.3 ^ABC^
P (mg/dL)	11.48 ± 0.69	11.62 ± 0.24 ^d^	11.70 ± 0.38	10.30 ± 0.24 ^b^	8.80 ± 0.19 ^BD^	10.98 ± 0.33 ^A^	9.95 ± 0.30 ^D^	10.90 ± 0.43 ^AC^
Ca (mmol/L)	10.96 ± 0.32 ^cd^	10.39 ± 0.28 ^cd^	8.06 ± 0.13 ^abd^	9.64 ± 0.18 ^abc^	10.54 ± 0.12 ^BCD^	9.42 ± 0.11 ^ACD^	7.48 ± 0.12 ^ABD^	8.68 ± 0.21 ^ABC^
OC (ng/mL)	10.97 ± 0.28 ^bc^	9.64 ± 0.33 ^ad^	9.05 ± 0.26 ^ad^	10.70 ± 0.14 ^bc^	10.38 ± 0.13 ^CD^	9.8 ± 0.13 ^D^	9.00 ± 0.29 ^AD^	11.68 ± 0.29 ^ABC^
CTX-I (ng/mL)	19.50 ± 0.35 ^bc^	21.55 ± 0.36 ^ac^	23.38 ± 0.39 ^ab^	19.45 ± 0.45 ^bc^	18.62 ± 0.95 ^CD^	19.22 ± 0.34 ^D^	21.55 ± 0.55 ^AD^	14.93 ± 0.83 ^ABC^

Explanations: as in Table 1. Abreviations: ALP—alcaline phosphatase; P—phosphorus; Ca—calcium; OC—osteocalcin; CTX-I—C-terminal telopeptides of type I collagen

## Data Availability

All data generated or analyzed during this study are included in this article.

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
