# Peer review of "Osteotropic Effect of Parenteral Obesity in Programmed Male Rats Fed a Calorically Differentiated Diet during Growth and Development"

_animals, 2022, doi:10.3390/ani12182314_

Round 1
Reviewer 1 Report
1. The rationale, aim, and hypothesis should be revisited and more clearly stated in the abstract and background portions of this manuscript. Right now, it is hard to determine what the gaps are in the literature and the importance of this study.
2. It is unclear why only male offspring were included in this study. Authors often made comparison to their previous female study (ref #16). Ref #16 was a male study. The authors also reference #14 as a comparison female study. I could not find/access ref #14, but the title suggests it is not a similar study design looking at the impact of diet programming on females, so, it is unclear why the authors are referring to it as such. The authors also referred to ref #15 as their female study, but this was a male study. The authors should avoid making direct comparisons between male and female responses to the diet because they did not study females.
3. Line 72, the same references are listed twice.
4. Line 75, regarding this statement, “There is also an opinion that obesity promotes the development of systemic inflammation.” Obesity is scientifically proven to cause low-grade systemic inflammation (PMID 28721154). Therefore, this statement is not an opinion. I advise this statement to be reworded for inclusion.
5. The authors should include the full (macronutrient and micronutrient) composition of the both diets in the methods section or in a supplemental file.
6. I see no rationale for the inclusion of circuiting lipids in the serum analysis. If this is due to the relationship between obesity and higher circulating lipid levels, then it should be included in the background section, and this should be linked to the expected effects on bone density. In addition, HDL, LDL, TG, and TC are all reported in Table 4. However, they are not mentioned anywhere in the results portion or in the discussion section. I believe that these data should be removed from this manuscript unless the author includes justification and explanation for its inclusion.
7. Table legends should make clear that if a letter “a, b, c, A”, etc. is present, that indicates a statistically significant difference.
8. ALL Tables 1-4. The tables should be rearranged into smaller, more digestible tables. For example, Table 1. Body weight, lean mass, fat mass, soft tissue mass. Table 2. Femur results Table 3. Whole skeleton results and so on.
9. ALL references. Please review all references and adjust them accordingly. After a thorough review, very few, if any, of the references reflect the statements they support. For example, authors cite reference #8 in support of a statement about the effects of leptin on bone. Ref #8 did not study leptin, but they cited a source for that statement. Authors should cite appropriate sources. Another example is that authors cited ref #12 in support of a statement about obesity causing inflammation, leading to bone resorption. Ref #12 does not have the word “obesity” or “inflammation” in it anywhere. These are just two examples; all references should be revisited. Additionally, the discussion section should include more outside references that do not belong to the authors. Many works are available on the effects of high-energy diets on bone. The manuscript should include some other references on different types of high-energy diets, to support or oppose, why you chose to use a high carbohydrate diet. More references overall should be incorporated to support your findings and provide gaps and details on what the author's study contributes to the field.
10. Line 299, there is no reference 29 in the reference list. There are only 24 references total, and none for said, Papageorgiou. Please review all references.
11. Line 178: Please clarify if standard error or standard deviation is presented. It says “standard error”, but is abbreviated as “SD” indicating standard deviation.
12. Line 264 states that the lowest concentration of osteocalcin was found in plasma of males in the S/S group. However, the table doesn’t reflect that. Also, the statistics for that claim state (P = 0.002 vs S/S). It is not clear what groups are being compared here since the claim is being made about the S/S group.
13. Line 266: (P = 0.01)…what is this statistic referring to? OC in the S/F group compared to what?
14. Can the authors comment as to why rats fed S/F weighed less than rats fed S/S? This seems counterintuitive. According to the authors, the F diet was fed for 90 days which is “the time necessary to induce obesity”. However, it did not appear to do so, given that the S/S rats weighed more than the S/F rats at days 49 and 90. Therefore, the conclusions made about the effects of “obesity” on bone are inaccurate since it doesn’t appear that the F diet induced obesity. At day 49, why does the S/S group have the highest body weight and adipose tissue mass? Isn’t that group expected to weigh the least?
Since the authors are making claims about the impact of “parental obesity” on the offspring, they should provide the body weight of the parents.
Author Response
First of all, the authors would like to underline their gratitude for the efforts of the Reviewer in the preparation of the revision of our manuscript. In response, we would like to present our rebuttal to the opinions expressed by the Reviewer.
Reviewer 1
We would like to express our sincere thanks to Reviewer 1 for their thorough analysis of the manuscript. For reasons that are not explained to us, EndNote incorrectly formatted the reference list, which prevented the correct interpretation of the content of the manuscript and its associated references. The currently cited references have been checked, corrected and supplemented, and we are convinced that this will allow the reviewer to understand the essence of the research undertaken. This also applies to the part of Discussion, where the numbering of the literature corresponds to the actual references cited and the problems discussed.
The double numbering in lines 72 has been corrected.
In line 75, the sentence containing "There is also an opinion that obesity promotes the development of systemic inflammation" has been corrected as suggested by the reviewer. A proposed reference was also cited.
The full composition of diets S and F has been added in supplementary materials as Table S1. As supporting materials, a figure of weight gain of female and male rats of the parental generation has also been included as Figure S1.
According to the reviewer's recommendations, the results concerning the lipid profile of the blood plasma were removed.
The reviewer's doubts are raised by the form of marking statistically significant differences using the letters "abcd" and "ABCD". However, the authors believe that it is clear and informative. The letters "abcd" are for males at 49 days of age, while the letters "ABCD" are the differences of the 90 day group. The explanation of this marking is detailed in the footnotes of each table and in Figure 2.
We would like to thank the reviewer for paying attention to the legibility of the tables with the results. Perhaps the current form of their presentation was less communicative for the reader, although tables with a large number of results dominate modern manuscripts. However, we consider creating a large number of small tables unjustified. Therefore, we propose a re-edited form of tables, and we believe that such a form is more "digestible".
The results presented are mean values and standard error of mean (SEM), not standard deviation (SD). Editing errors have been corrected.
In line 264, the concentration of OC concerned the F/S group and not the S/S group, and the statistically significant difference was related to the F/S and S/S groups. The correction has been made.
Line 266 was corrected and the description was made more specific
The reviewer rightly noticed the lack of male body weight gain in the F/F group. Thank you very much for drawing attention to this oversight. The person developing the data tables made an editing error entering the wrong value of the male body weight in the F/F_90 group. It should be 620.50 g and this value was significantly higher (P < 0.05) compared to the S/S_90 group. The reviewer's doubts are raised by the fact that males in the F/F_49 group are lower than in the S/S_49 groups. Also, for us, this relationship was surprising and puzzling. We want to add that a similar dependence was also noted in females (date not published). The authors admit that it is difficult to explain this clearly. We believe, however, that this should be related to the fact that puberty is a time of multidirectional changes in the body's physiological processes, which also affect body weight. It should be emphasized that this dependence only applies to the period from 21 to 49 days of age. In the next period (49-90 days of age), the differences in body weight are eliminated, and the body weight of F / F_90 males is significantly greater compared to the S / S group. The consistency of the results of body mass and body composition parameters deserves to be emphasized.
Ultimately, the authors decided to remove the bone mass and length results as not mentioned in the manuscript.
The authors have a hope that the Reviewer will be fully satisfied.

Reviewer 2 Report
In the manuscript entitled: 'Osteotropic effect of parenteral obesity in programmed male rats fed a calorically differentiated diet during growth and development' Radzki et al. present data on the effect of diet on skeletal strength in prenatally programmed rats. The experiments are well designed and the article is well-written.
I suggest to expand the discussion and conclusion. Authors may want to discuss their interesting finding that the prenatally programmed bone responds to dietary modification post-weaning.
Author Response
First of all, the authors would like to underline their gratitude for the efforts of the Reviewer in the preparation of the revision of our manuscript. In response, we would like to present our rebuttal to the opinions expressed by the Reviewer.
Reviewer 2
Thank you very much for your review. As suggested by the reviewer, the authors supplemented the information in the discussion and the literature. Supplementary data such as the full composition of diets S and F as Table S1 and a figure of weight gain of female and male rats of the parental generation as Figure S1 have also been added.

Round 2
Reviewer 1 Report
Since authors removed bone length from results, any mention of bone length (see last paragraph of discussion section) should be removed.
Author Response
Thank you very much to the reviewer for re-reviewing the manuscript. According to the reviewer's comments, the last paragraph of the manuscript was corrected. The description concerning the length of the isolated femur has been replaced with data describing its area.